# Tension activation of mechanosensitive two-pore domain K+ channels TRAAK, TREK-1, and TREK-2

Ben Sorum[1,2,3,4,5], Trevor Docter[1,2,3,5], Vincent Panico[1,2,3], Robert A. Rietmeijer[1,2,3] & Stephen G. Brohawn [1,2,3] ✉

TRAAK, TREK-1, and TREK-2 are mechanosensitive two-pore domain K+ (K2P) channels that contribute to action potential propagation, sensory transduction, and muscle contraction. While structural and functional studies have led to models that explain their mechanosensitivity, we lack a quantitative understanding of channel activation by membrane tension. Here, we define the tension response of mechanosensitive K2Ps using patch-clamp recording and imaging. All are low-threshold mechanosensitive channels ($T_{10\%/50\%}$ 0.6-2.7 / 4.4-6.4 mN/m) with distinct response profiles. TRAAK is most sensitive, TREK-1 intermediate, and TREK-2 least sensitive. TRAAK and TREK-1 are activated broadly over a range encompassing nearly all physiologically relevant tensions. TREK-2, in contrast, activates over a narrower range like mechanosensitive channels Piezo1, MscS, and MscL. We further show that low-frequency, low-intensity focused ultrasound increases membrane tension to activate TRAAK and MscS. This work provides insight into tension gating of mechanosensitive K2Ps relevant to understanding their physiological roles and potential applications for ultrasonic neuromodulation.

Mechanosensitive ion channels are opened by mechanical force to rapidly transduce physical stimuli into cellular electrical signals[1–3]. Their activity underlies a wide range of physiological processes, from the classic senses of touch and hearing to proprioception, blood pressure regulation, digestion, osmolarity control, and cell growth and division. Known mechanosensitive channels are diverse. They belong to evolutionarily distinct families and exhibit varying ion selectivity, kinetics, conductance, force-gating mechanisms, and sensitivity to mechanical stimuli[1,2]. Characterization of these properties is essential for understanding how forces are sensed and encoded, but force activation remains to be quantitatively described for many mechanosensitive channels.

TRAAK, TREK-1, and TREK-2 are mechanosensitive members of the two-pore domain (K2P) K+ ion channel family[4]. They display low

basal leak activity under resting conditions, but are activated up to ~100-fold by increased membrane tension[5,6]. Leak and mechanically gated activity of TRAAK arise from physically distinct open states[5]. At low tension, TRAAK is predominantly closed due to lipid block of the pore[7]. Delipidation produces leak activity, while mechanically gated activity involves conformational changes—most notably the "upward" movement of transmembrane helix 4 towards the extracellular solution—that seal laterally facing membrane openings to prevent lipid block of the ion-conducting pore[5,7]. Similar conformational changes have been observed in structures of TREK-2[8]. These conformational changes increase channel cross-sectional area and cylindricity, transformations that are energetically favored by increased tension[7].

Whether and how mechanical gating of TRAAK, TREK-1, and TREK-2 contributes to their physiological functions is generally unknown.

[1]Department of Molecular & Cell Biology, University of California Berkeley, Berkeley, CA 94720, USA. [2]Helen Wills Neuroscience Institute, University of California Berkeley, Berkeley, CA 94720, USA. [3]California Institute for Quantitative Biosciences (QB3), University of California, Berkeley, CA 94720, USA. [4]Present address: Department of Biomedical Sciences, Cooper Medical School of Rowan University, Camden, NJ 08103, USA. [5]These authors contributed equally: Ben Sorum, Trevor Docter. ✉e-mail: brohawn@berkeley.edu

TRAAK and TREK-1 are localized to nodes of Ranvier, the small gaps between myelinated regions of axons where the action potential is regenerated during saltatory conduction[9–11]. In contrast to the classically studied squid giant axon, from which the canonical model of the molecular basis for the action potential was derived, mammalian axons can completely lack voltage-gated K⁺ channels at nodes[12]. Instead, TRAAK and TREK-1 contribute to the large nodal leak K⁺ conductance that sets the resting potential, repolarizes the membrane in an action potential, and maintains voltage-gated Na⁺ channel availability to facilitate high-frequency spiking[9–11]. Consistent with an important physiological role, gain-of-function mutations of TRAAK in humans cause the severe neurodevelopmental disorder FHEIG (facial dysmorphism, hypertrichosis, epilepsy, intellectual disability/developmental delay, and gingival hyperplasia) and epilepsy[13,14]. While TRAAK expression is apparently restricted to nodes of Ranvier, TREK-1 and TREK-2 are expressed more broadly in the nervous system and in other tissues, including the heart and smooth muscle[15].

Membrane tension is the known or suspected gating stimulus for many mechanosensitive ion channels, but it is not typically measured in assays of channel activity[3,16]. Instead, stimulation parameters including probe displacement during cell poking, substrate elongation or pillar displacement during cell stretching, osmolarity during cell swelling, or applied pressure during patched membrane stretching are measured and reported. These stimuli are all thought to increase membrane tension in addition to other assay-specific effects. Relating measured parameters to tension is not trivial. One solution is to use electrophysiology to record currents and simultaneously image patched membranes during pressure-induced activation of mechanosensitive channels[17–26]. Membrane tension ($T$) can then be calculated according to the Young-Laplace equation ($T = \Delta Pr/2$) using values of applied pressure ($\Delta P$) and patch membrane radius of curvature (r) determined from images. This approach has been used to characterize the tension response of Piezo1, MscS, and MscL[17–26].

An emerging approach for activating mechanosensitive channels is low-power and low-frequency ultrasound stimulation[27,28]. Activation of mechanosensitive ion channels may underlie ultrasound's neuromodulatory properties[6,29–32] that were first identified nearly a century ago and have been widely studied throughout the nervous system since. Ultrasound is advantageous for neuromodulation and manipulating ion channel activity as it can be focused and delivered noninvasively through tissue and bone[27,28]. We previously showed that ultrasound activates TRAAK in patches from cells and proteoliposomes and promotes the same mechanically gated open state as pressure stimulation[6]. This suggests ultrasound can activate mechanosensitive channels by increasing membrane tension rather than through alternative means of energy transfer, but this has not been shown definitively.

Here, we use simultaneous patch imaging and recording to quantify the tension response of TRAAK, TREK-1, and TREK-2. We further show that ultrasound stimulation stretches patched membranes consistent with tension-mediated activation of mechanosensitive channels, including TRAAK and MscS.

## Results

We expressed human TRAAK, TREK-1, or TREK-2 channels in *Xenopus laevis* oocytes and recorded currents across excised (inside-out) patches in response to mechanical stimulation generated by pressure steps to the base of the patch pipette (Fig. 1A, C, E). Currents were recorded at 0 mV in a tenfold gradient of K⁺ across the membrane. Patches contained hundreds of channels given single-channel currents of ~1–2 pA under these conditions[5,15]. For each patch, the peak current during a pressure step (I) was normalized to the maximum current elicited by mechanical stimulation ($I_{max}$) so data from patches with different numbers of channels could be compared (Fig. 1B, D, F). Patches were only analyzed if a saturating current response to pressure was observed. For all three channels, activity was low in resting

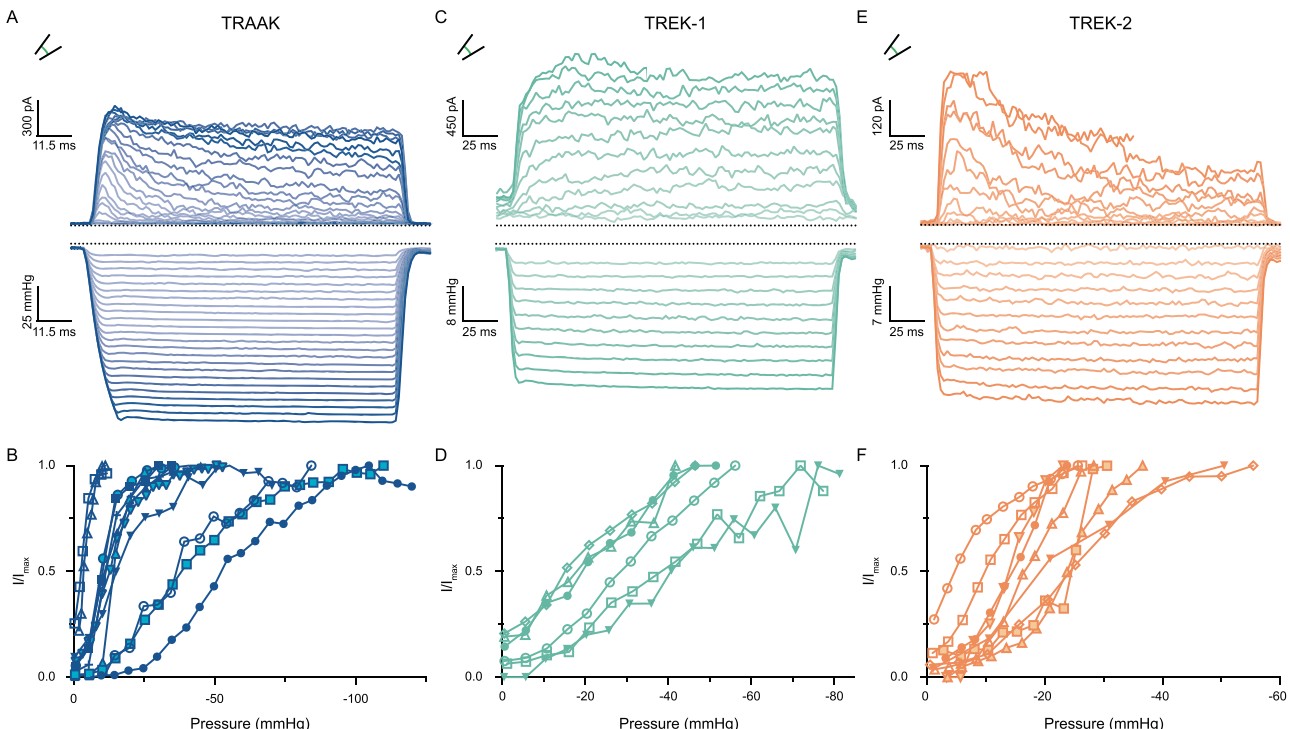

**Fig. 1 | Pressure stimulation activates TRAAK, TREK-1, and TREK-2 channels.**
**A, C, E** Macroscopic currents (upper) recorded at 0 mV in a tenfold [K⁺] gradient from a **A** TRAAK-, **C** TREK-1-, or **E** TREK-2-containing patch in response to a pressure step protocol (lower). **B, D, F** Normalized current–pressure relationships from **B** 12 TRAAK-, **D** 6 TREK-1-, or **F** 9 TREK-2-containing patches (from *n* = 5, 3, and 6 cells, respectively). Normalized current is defined as the peak current recorded during each pressure step (I) over the maximum current recorded for each patch ($I_{max}$). Data from each patch are shown as different-shaped points connected by a line.

membranes and highly activated by increased pressure, consistent with prior reports[5,6]. For TRAAK, TREK-1, and TREK-2, pressure increased activity a maximum of $41.2 \pm 14.2$-fold, $9.5 \pm 1.8$-fold, and $14.1 \pm 2.9$-fold, respectively (mean $\pm$ SEM, $n = 12$, 6, and 9 patches).

We compared the mechanical activation of TRAAK, TREK-1, and TREK-2 to the well-characterized *E. coli* mechanosensitive channel MscS[33] (Supplementary Fig. 1A, B). MscS was activated by negative pressures up to an average of $14.6 \pm 4.0$-fold (mean $\pm$ SEM, $n = 8$ patches). Patches contained tens of channels given a single-channel current of -12 pA at −60 mV under these conditions[34]. In all cases (Fig. 1B, D, E and Supplementary Fig. 1B), patches showed substantial variability in their current response to pressure. This is consistent with membrane tension, rather than pressure, being the stimulus that promotes channel opening[23].

Membrane tension ($T$) is related to the pressure difference across the patch ($\Delta P$) and the membrane radius of curvature (r) according to the Young-Laplace equation: $T = \Delta Pr/2$. To calculate tension during mechanical stimulation, we visualized the membrane during pressure steps and determined the radius of patch curvature. TRAAK, TREK-1, and TREK-2 were co-expressed with plasma membrane-targeted EGFP (EGFP fused to the CAAX lipidation motif from H-Ras) and MscS was directly fused to EGFP. EGFP fluorescence in patched membranes was imaged at a 120 Hz frame rate during recordings. Pressure-induced changes in membrane curvature were readily visible in patches, with the magnitude of curvature change increasing with increasing pressure (Fig. 2A and Supplementary Fig. 2). Movie frames corresponding to the peak current elicited during each pressure step were selected for analysis and patch radius of curvature was determined by fitting a circle to points on the membrane identified by fluorescence. The resulting radii and measured pressure were used to calculate tension. Global fits to a Boltzmann function showed consistent channel response to tension across patches (fit $R^2 = 0.83$, 0.70, 0.88, and 0.82 for MscS, TRAAK, TREK-1, and TREK-2, respectively (Supplementary Fig. 1C and Fig. 2B–D)). We found MscS was activated with a midpoint $T_{50} = 3.7 \pm 0.2$ mN/m, slope factor of $1.1 \pm 0.2$ mN/m, and 10–90% activation range of 1.2–6.3 mN/m (mean $\pm$ SEM, $n = 4$ patches) (Supplementary Fig. 1C), consistent with prior studies[21].

The mechanosensitive K2Ps differed in their tension sensitivity and tuning. TRAAK was activated over a broad range of tension with a midpoint $T_{50} = 4.4 \pm 0.2$ mN/m, slope factor $1.7 \pm 0.2$ mN/m, and 10–90% activation range 0.6–8.2 mN/m (mean $\pm$ SEM, $n = 12$ patches). The TREK-1 response was broader and right shifted to higher tension by ~2 mN/m ($T_{50} = 6.4 \pm 0.2$ mN/m, slope factor $2.3 \pm 0.2$ mN/m, and 10–90% activation range 1.5–11.3 mN/m (mean $\pm$ SEM, $n = 6$ patches)). TREK-2 has an intermediate $T_{50}$ and the narrowest response range among the K2Ps ($T_{50} = 5.8 \pm 0.1$ mN/m, slope factor $1.4 \pm 0.1$ mN/m, and 10–90% activation range 2.7–8.9 mN/m (mean $\pm$ SEM, $n = 9$ patches)). Comparing the tension response of TRAAK, TREK-1, TREK-2, and MscS shows that the channels all have low thresholds for activation (defined by the 10% activation tension from Boltzmann fits). However, TRAAK and TREK-1 respond over a much broader range of tension than TREK-2 and MscS (Fig. 2D). In other words, the MscS and TREK-2 responses are steep and switch-like, while TRAAK and TREK-1 responses are more graded.

We next assessed ultrasound stimulation of TRAAK and MscS using the same patch imaging and recording setup. We designed a 3D-printed recording chamber to isolate the mechanical effects of

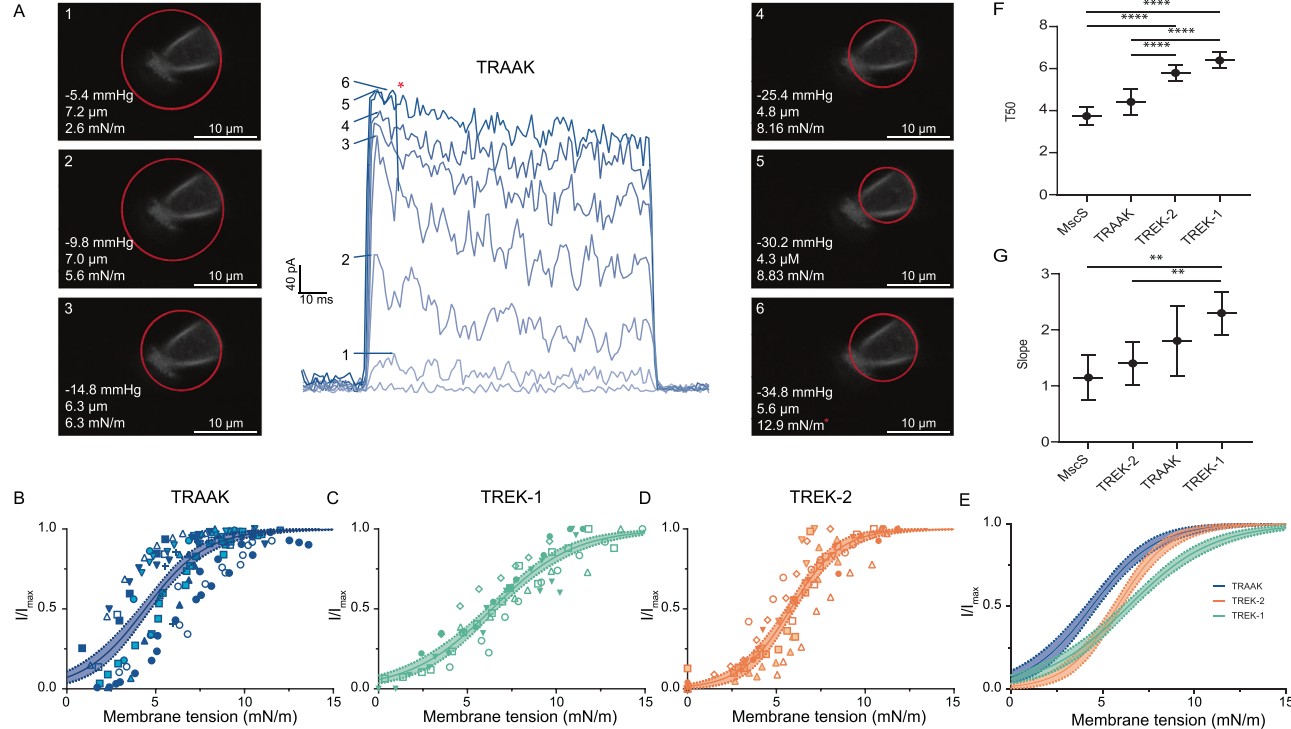

**Fig. 2 | Quantification of TRAAK, TREK-1, and TREK-2 activation by membrane tension. A** Macroscopic current (center) was recorded at 0 mV in a tenfold [K$^+$] gradient from a TRAAK-containing patch in response to a pressure step protocol. Fluorescent images of the patched membrane at the time of maximum current response during each of the six pressure steps are shown. Measured pressure, patch radius (from a red fit circle), and calculated tensions are shown in each image. The asterisk indicates the point just prior to patch rupture. Normalized current-tension relationships for **B** TRAAK, **C** TREK-1, and **D** TREK-2. Global fits to a Boltzmann sigmoidal with 95% confidence intervals are shown. TRAAK $T_{50} = 4.4 \pm 0.2$ mN/m, TREK-1 $T_{50} = 6.4 \pm 0.2$ mN/m, and TREK-2 $T_{50} = 5.8 \pm 0.1$ mN/m, mean $\pm$ SEM for $n = 12$, 6, and 9 patches (from $n = 5$, 3, and 6 cells), respectively. Data from each patch are shown as differently shaped or shaded points. **E** Overlaid fits comparing tension response of mechanosensitive K2Ps. Comparison of **F** $T_{50}$ and **G** slope factor from TRAAK, TREK-1, TREK-2, and MscS fits (mean $\pm$ SEM for $n = 12$, 6, 9, and 4 patches (from $n = 5$, 3, 6, and 3 cells), respectively. Differences were assessed with one-way analysis of variance (ANOVA) with Dunnett correction for multiple comparisons, ****$p < 0.0001$, **$p < 0.01$).

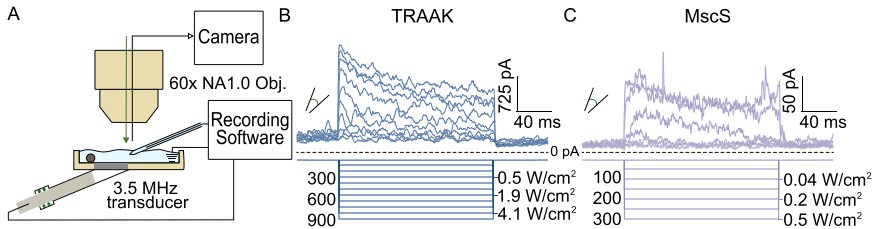

**Fig. 3 | Ultrasound stimulation activates TRAAK and MscS channels.**
**A** Schematic of recording setup for ultrasound stimulation during simultaneous patch recording and imaging. A 3.5 MHz ultrasound transducer is mounted into a custom 3D-printed recording dish and patched membranes are positioned at the position of maximum ultrasonic power. **B** Macroscopic currents from a TRAAK-containing patch in response to an ultrasound step protocol ($V_{hold} = 0$ mV,

transducer driving voltage = 0–1 V, Δ driving voltage = 100 mV. 100 mV intervals displayed with measured power at patch position in W/cm² indicated). **C** Macroscopic currents from a MscS-containing patch in response to an ultrasound step protocol ($V_{hold} = 0$ mV, transducer driving voltage = 0–0.3 V, Δ driving voltage = 50 mV, measured power at patch position in W/cm² indicated).

ultrasound and ensure consistent ultrasound intensity at the patch (Fig. 3A). A 3.5 MHz ultrasound transducer was connected to the chamber through a mylar partition and patched membranes were positioned at the point of maximum ultrasonic intensity. Channels were stimulated with 200 ms ultrasound bursts of increasing power from 0.01 to 5.4 W/cm². As in our previous work[6], we designed stimulation protocols to minimize bath temperature increases (to less than an estimated 0.05 °C) to exclude potential thermal activation of channels.

Increasing steps of ultrasound power increasingly activated TRAAK (Fig. 3B and Supplementary Fig. 3A–D) with a midpoint power of $1.8 \pm 0.07$ W/cm². At the highest ultrasound intensities achieved before patch rupture, TRAAK was activated $16.2 \pm 4.2$-fold (mean ± SEM, $n = 17$ patches). Lower frequency ultrasound stimulation (2.25 MHz) resulted in comparable TRAAK activation up to $19.3 \pm 4.2$-fold with a midpoint power of $0.38 \pm 0.01$ W/cm² (mean ± SEM, $n = 6$ patches). MscS was similarly activated by 3.5 MHz ultrasound stimulation up to $13.0 \pm 2.0$-fold and with a midpoint power of $0.15 \pm 0.02$ W/cm² (mean ± SEM, $n = 8$ patches) (Fig. 3C and Supplementary Fig. 3E, F). The channels' ultrasound responses mirror their tension responses; TRAAK and MscS have similarly low thresholds for activation, but TRAAK shows a broader response range and higher midpoint power for activation.

Patch imaging showed that ultrasound stimulation, like pressure, induced membrane curvature changes that were concomitant with channel activation (Fig. 4A). In most patches, the membrane deflected outwards towards the pipette tip during ultrasound stimulation as observed for positive pressure application. TRAAK and MscS activation was similar when membrane tension was generated by positive or negative pressure stimulation that resulted in opposite membrane curvature (Supplementary Fig. 4), consistent with previous reports for TRAAK[35,36]. The similarity of stimulus-induced curvature changes suggests that both ultrasound and pressure increase membrane tension to activate mechanosensitive TRAAK and MscS channels. We reasoned that if this is the mechanism for ultrasonic channel activation, then the membrane curvature of a patch should be the same when channels are activated to the same degree by pressure or ultrasound. Indeed, patch radii were indistinguishable when TRAAK was comparably activated by either pressure or ultrasound (Fig. 4A–C, $n = 9$ paired recordings from 5 patches, $p = 0.18$, paired $t$-test). Analysis of MscS gave the same result (Supplementary Fig. 5, $n = 7$ paired recordings from 5 patches, $p = 0.95$, paired $t$-test). Control comparisons of paired records from the same patch that yield comparable TRAAK activation show that (1) paired pressure stimuli result in indistinguishable patch radii and membrane tension (Supplementary Fig. 6A, $n = 14$ paired recordings from 5 patches, $p = 0.79$, two-tailed paired $t$-test), (2) paired ultrasound stimuli result in indistinguishable patch radii (Supplementary Fig. 6B, $n = 6$ paired recordings from 4 patches, $p = 0.47$, two-tailed paired $t$-test), and (3) comparable patch radii are observed regardless of the order in which pressure and ultrasound stimuli are presented (Supplementary

Fig. 6C, $n = 3$ paired recordings from two patches, $p = 0.97$, two-tailed paired $t$-test). Together, these results are consistent with ultrasound increasing membrane tension to activate mechanosensitive channels like canonical mechanical stimuli.

## Discussion

In this study, we used simultaneous patch imaging and recording to quantify the full tension response of the mechanosensitive K⁺ channels TRAAK, TREK-1, and TREK-2. We note that there are several caveats intrinsic to characterizing tension sensitivity in excised patches. First, we assume uniform tension across the patched membrane. Second, tension calculation requires hemispherical membranes (in perfectly flat patches with infinite curvature, the Laplace–Young equation is undefined) and it becomes increasingly challenging to calculate low tension values as patch radii increase in flatter membranes[37]. The lowest tension we measure is 0.36 mN/m. Third, membranes can have significant and variable basal tensions prior to stimulation due to lipid adhesion to glass, estimated to be ~0.5–4 mN/m in one study[38]. We attempted to minimize variability by pulling patches with low basal curvature and low resting channel activity consistent with low basal tension (see Methods)[5,6], but some variability in resting tension is present between patches and this is unaccounted for in our calculations of stimulus-induced tension. Still, our results for MscS are consistent with prior studies, supporting the reliability of our approach[21]. In addition, our results for TRAAK are consistent with a study of TRAAK activation by low- to mid-tensions (0.8 to 5.7 mN/m) in planar lipid bilayers[36].

We show that TRAAK, TREK-1, and TREK-2, like MscS, MscL, and Piezo1, are membrane tension-gated[3,16] and compare the tension response of mechanosensitive K2Ps to other channels (Fig. 5). *E. coli* MscS has a low $T_{50}$ of ~4 mN/m measured in patches from cells or ~2.5–7 mN/m from reconstituted proteoliposomes depending on lipid composition[17–21]. *E. coli* MscL has a high $T_{50}$ of 11–12 mN/m measured from proteoliposomes or cell membranes[22–24]. Both bacterial channels show a steep tension response over a narrow range. The switch-like opening is likely to be essential for the protective role of MscS and MscL as pressure-release valves in the bacterial response to osmotic shock[3,33]. Piezo1 is as or more sensitive than MscS with a $T_{50}$ of ~1.5–3 mN/m or ~5 mN/m measured in on-cell patches or excised cell patches, respectively[24–26]. Piezo1, like MscS and MscL, responds over a narrow range of tension with a step-like response to function like a switch to depolarize cells in response to mechanical force in numerous physiological contexts[1,2]. TRAAK and TREK-1 are similarly sensitive to Piezo1 and MscS, with a threshold of ~1 mN/m, but show a notably broader response and correspondingly higher $T_{50}$s of ~4.5 and 6.5 mN/m, respectively. TRAAK and TREK-1 therefore can provide a graded K⁺ conductance proportional to mechanical force magnitude across nearly the entire range of biologically feasible tension. Within a node of Ranvier, tension modulation of TRAAK and TREK-1

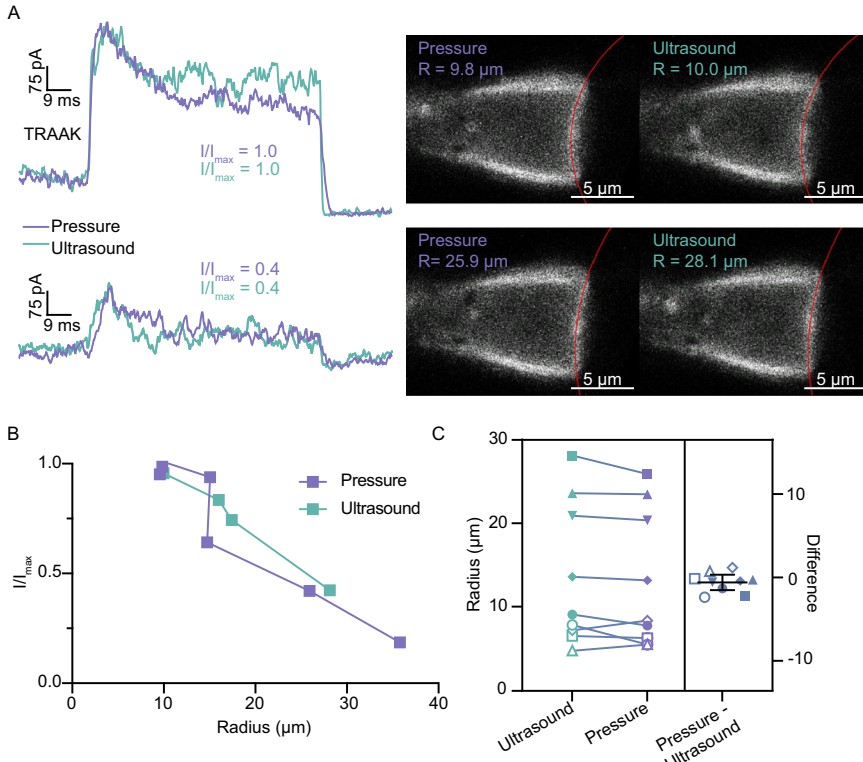

**Fig. 4 | Ultrasound and pressure generate membrane tension to activate TRAAK channels. A** Overlaid macroscopic currents (left) and corresponding fluorescent images (right) from a TRAAK-containing patch in response to pressure (periwinkle) and ultrasound (green) stimulation. Traces and images are compared from the same patch activated to high (upper) and moderate (lower) $I/I_{max}$. Circle fits (red) and corresponding membrane radii of curvature are indicated on the images. Similar patch radii are observed when ultrasound or pressure stimuli generate similar channel activation. **B** Normalized current-patch radius relationship calculated from a single patch in response to ultrasound and pressure stimulation. **C** Comparison of patch radius generated by pairs of stimuli (with pressure preceding ultrasound) that give comparable channel activation in the same patch. Data from nine paired recordings from five different patches are shown ($p = 0.18$, two-tailed paired $t$-test, not significant).

activity could impact axonal excitability through, for example, altered repolarization rate, resting potential, input resistance, or voltage-gated sodium channel availability. In contrast, TREK-2 responds with an intermediate $T_{50}$ of ~6 mN/m over a steeper range.

In the presence of membrane tension, expansion of protein cross-sectional area ($\Delta A$) is favored by an energy equal to $-T\Delta A$. If we assume area expansion is the dominant term driving the opening of a mechanosensitive channel, $\Delta A$ and the intrinsic energy difference ($\Delta G$) between an open and closed state can be derived from fitting tension response to the Boltzmann equation $P_O = 1/1+\exp((\Delta G - T\Delta A)/k_BT)$[37,39]. We calculate for TRAAK $\Delta A = 2.4$ nm$^2$ and $\Delta G = 10.5 \times 10^{-21}$ J (2.6 k$_B$T), for TREK-1 $\Delta A = 1.8$ nm$^2$ and $\Delta G = 11.4 \times 10^{-21}$ J (2.8 k$_B$T), and for TREK-2 $\Delta A = 2.9$ nm$^2$ and $\Delta G = 17.0 \times 10^{-21}$ J (4.1 k$_B$T). The area change for TRAAK derived from electrophysiological recordings (2.4 nm$^2$) is nearly identical to the maximal area change calculated from experimental structures captured in TM4 down and mechanically-activated TM4 up states (2.7 nm$^2$)[5,7]. This provides further support for a model in which mechanical gating involves movement of transmembrane helix 4 upward, expanding channel cross-sectional area and sealing membrane-facing openings to lipid block[5,7].

We used the patch imaging and recording setup developed here to gain insight into the basis of ultrasound activation of mechanosensitive channels. Among neuromodulatory techniques, ultrasound is uniquely focusable and penetrant in biological tissues. This means ultrasound can, for example, be targeted non-invasively to deep brain structures through the skull and elicit excitatory or inhibitory effects on neural activity that depend on the target and stimulus parameters[27,28]. Recent studies have implicated mechanosensitive channels as mediators of some ultrasound effects, with channel activation observed in vitro,

upon heterologous expression, and from endogenously-expressed channels in central and peripheral neurons[6,29–32,40–45]. Harnessing and predicting ultrasound activation of endogenous or exogenously expressed channels will require a more complete understanding of the underlying molecular mechanisms involved.

We demonstrated both TRAAK and MscS channels are activated by ultrasound in excised patches. We observed similar response profiles to ultrasound and pressure stimulation for each channel and found that both stimuli generate changes in patch curvature correlated with channel activation. Within a single patch, ultrasound and pressure stimulation are indistinguishable in generating a relationship between membrane radius of curvature and channel activity, suggesting both stimuli activate channels through membrane tension. This is consistent with our previous work showing ultrasound and pressure promote the same mechanically gated TRAAK open state[6]. Together, these results suggest ultrasound activates mechanosensitive channels by increasing membrane tension. Other potential effects, including temperature increase, cavitation, or acoustic scattering, may be relevant under some conditions[27,28], but are not likely to explain channel activation by low-intensity ultrasound. Instead, acoustic radiation force and resulting acoustic streaming likely increase membrane tension to mechanically activate channels[27,40,42]. Consistently, work in other systems has demonstrated mechanical displacements of reconstituted lipid bilayers and cell membranes by ultrasound[46,47].

Ultrasound has been shown to produce both inhibitory and excitatory effects on neuronal activity depending on the system and stimulation parameters used[27,32]. Since TRAAK and TREK-1 are localized to nodes of Ranvier in myelinated axons where they contribute to spike propagation[9,10], it is possible their activation by ultrasound underlies

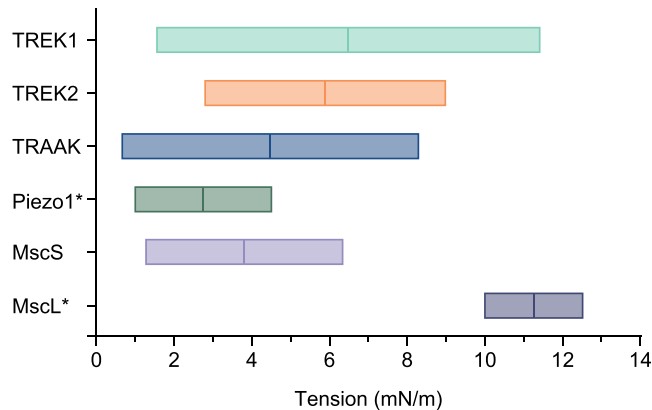

**Fig. 5 | Comparison of mechanosensitive channel tension response ranges.**
Tension response ranges of mechanosensitive channels TRAAK, TREK-1, TREK-2, MscS (from this study), Piezo1 (from ref. 25), and MscL (from ref. 23). Boxes encompass the 10–90% activation range. $T_{50}$ values are indicated by vertical lines.

some of these effects. Increased nodal potassium conductance could conceivably inhibit spiking by hyperpolarizing the membrane and decreasing input resistance or facilitate spiking under conditions of limited voltage-gated sodium channel availability by promoting their recovery from inactivation. Future work that combines targeted ultrasound stimulation to nerves or white matter with genetic manipulation of TRAAK and TREK-1 could shed light on their contribution to ultrasonic neuromodulation.

Sonogentic approaches have sought to sensitize cells to ultrasound through heterologous expression of ultrasound-sensitive mechanosensitive channels, including Piezo1[40,41], TRAAK[6], and tension-sensitized MscL mutants (G22S[43–45] or I92L[42]). Our results suggest that ultrasound stimulation increases membrane tension to activate these channels and that efforts to engineer channels with lower tension activation thresholds or narrow tension responsive ranges could yield improved sonogentic tools. A wide range of ultrasound stimulation parameters have been reported to activate mechanosensitive channels with varying frequencies (from 300 kHz to 42 MHz), powers (0.05 to 750 w/cm$^2$), beam profiles (with foci of 0.2 to 4 mm$^2$), and waveforms (e.g., durations from 10–200 ms) producing varying effects[6,28–31,40–42]. We observe modest differences in midpoint powers for TRAAK activation by different frequencies used here and in our prior study[6] (0.38 W/cm$^2$ at 2.25 MHz, 1.82 W/cm$^2$ at 3.5 MHz, and 0.8 W/cm$^2$ at 5 MHz). This may be due in part to differences in the efficacy of tension generation in patches by different stimulation frequencies. We note, however, that midpoint powers may be underestimated because it was not always possible to verify maximal ultrasonic TRAAK activation in these experiments with saturating current responses prior to patch loss. Still, identifying stimulation protocols that maximally increase membrane tension could improve the efficacy of ultrasonic neuromodulation and sonogenetics.

## Methods

### Ethics statement
Animal procedures were reviewed and approved by the Animal Care and Use Committee at the University of California, Berkeley (AUP 2019-11-12743-1).

### Expression in and recording from *Xenopus laevis* oocytes. 
Genes encoding full-length *Homo sapiens* TRAAK (UniProt Q9NYG8-2), TREK-1 (UniProt O95069), and TREK-2 (UniProt P57789) were codon optimized for eukaryotic expression (without changing the native amino acid sequence), synthesized (Genewiz), and cloned into a modified pGEMHE vector using Xho1 and EcoR1 restriction sites. The transcribed messages encode *H. sapiens* TRAAK amino acids 1–393, TREK-1 amino acids 1–426,

or TREK-2 amino acids 1–538 with an additional three amino acids (SNS) at the C terminus. The coding sequence for *E. coli* MscS[34,48] amino acids 1–286 with N-terminal FLAG and GFP tags was cloned into the same modified pGEMHE vector using EcoR1 and Xho1 restriction sites. The transcribed message encodes pFLAG-CTC-GFP-MscS-SNS[48]. A construct encoding EGFP fused to the CAAX-containing C-terminal tail of *H. sapiens* H-Ras (NP_005334 amino acids 170–189) through a GGRS linker was cloned into a pCS2+ vector with a CACC Kozak sequence using Gibson assembly. Linearized DNA was transcribed in vitro using T7 polymerase. Comlementary RNA (0.1 to 10 ng for TRAAK, TREK-1, TREK-2, and MscS and 3–10 ng for EGFP-CAAX) in 50 nL $H_2O$ was injected into *Xenopus laevis* oocytes extracted from anesthetized frogs. Currents were recorded at 25 °C from inside-out patches excised from oocytes 1 to 5 d after RNA injection. The pipette solution contained 15 mM KCl, 135 mM NaCl, 2 mM $MgCl_2$, and 10 mM HEPES (pH = 7.4 with KOH) and the bath solution contained 150 mM KCl, 2 mM $MgCl_2$, 1 mM EGTA, 10 mM HEPES (pH = 7.1 with KOH). Currents were recorded using an Axopatch 200B Patch Clamp amplifier at a bandwidth of 1 kHz and digitized with an Axon Digidata 1550B at 500 kHz. Pressure was applied with a second-generation high-speed pressure clamp device (HSPC-2-SB, ALA Scientific Instruments). Aggregated data were from patches from $n \geq 2$ different cells on different days. No relevant differences were observed between patches from different cells. We attempted to minimize variability in resting tension by pulling patches with low basal curvature and low resting channel activity consistent with low basal tension. Pipettes pulled with short taper and moderate diameter tips (with a corresponding resistance of 2–3 MΩ in recording solutions) resulted in a higher proportion of patches with low basal activity. This also reduced the frequency of run-up of basal activity (likely due to patch creep up the pipette wall and increased tension[49]) after repeated mechanical stimuli. Patches with basal activity corresponding to $I/I_{max} > 0.25$ were not analyzed.

**Ultrasound setup and application.** We conducted both pressure and ultrasound recordings in a custom 3D-printed mount that placed the ultrasound transducer in line with the patch pipette at a 20° angle. The recording chamber, containing the ultrasound transducer mount and recording bath, was made from a clear SLA photopolymer (Formlabs RS-F2-GPCL-04). To ensure the bath fluid would not leak out of the chamber, the transducer mount was fitted with two nitrile O-rings (SUR&R 55AA89). Inside-out patches were excised from oocytes within the ultrasound chamber. The patch was centrally positioned ~1 in (25.4 mm) away from the cylindrical transducer surface, separated by bath solution and a thin sheet of mylar which the oocyte rested upon. An ultrasound wave was generated using a V326-SU (Olympus) focused-immersion ultrasonic transducer with a 9.525 mm nominal element diameter, 25.2 mm focal length, and an output center frequency of 3.5 MHz. For (Fig. S3A, B), an ultrasound wave was generated using a V325 (Olympus) focused-immersion ultrasonic transducer with a 9.525 mm nominal element diameter, 34.3 mm focal point, and an output center frequency of 2.25 MHz. To trigger ultrasound pulses, a function generator (Agilent Technologies, model 33220 A) was used to send an input voltage waveform to an ENI RF (radio frequency) amplifier (model 403LA), which provided output voltage to the ultrasound transducer for producing the stimulus waveform. The timing of the ultrasound stimuli was controlled by triggering the function generator manually or by software (Clampex 10.7). In the case of ultrasound pulse generation through software, a Clampex 10.7-generated waveform triggered a first function generator through a digitizer (Axon Digidata 1550B), which triggered a second function generator, which triggered the RF amplifier that drives the ultrasound transducer. Solutions were degassed to minimize microbubble cavitation and ultrasound attenuation.

**Calculating ultrasound pressure and power.** The output pressures were measured using a calibrated hydrophone (Onda, model HNR-0500).

The hydrophone measurements were performed at the position of peak pressure. When converting the measured voltages into pressures, we accounted for the hydrophone capacitance according to the manufacturer's calibration. Using the appropriate conversion factor listed under the Pascals-per-volt column on the look-up table that was supplied with the calibrated hydrophone, the hydrophone voltage-trace waveform was transformed into an acoustic-pressure waveform in MPa. We calculated the ultrasound power intensity in Watts/square centimeter (W/cm²) with the following equation:

$$I = \frac{p^2}{Z} = \frac{(P \times 0.707)^2}{\left(1.48 \times 10^6 \frac{kg}{m^2 s}\right)} \left(\frac{1}{100^2}\right) \tag{1}$$

**Patch imaging and membrane tension calculation.** Excised patches were illuminated with an LED light engine (SpectraX, Lumencor) through a GFP filter (450/50 nm excitation, 506 nm dichroic mirror, 500 nm longpass emission filter) and water immersion objective lens (x60, NA1.0). Movies were recorded at 120 Hz with an infrared camera (IR-2000, DAGE-MTI). Images were analyzed within FIJI (ImageJ). Image contrast was enhanced to facilitate analysis. Video files were loaded into FIJI and converted into a JPEG stack. Frames were then time-matched to stimuli by multiplying frame rate and time. Frames of interest corresponding to the maximum current during each pressure step were analyzed using two methods (available at [https://github.com/BrohawnLab/TensionScripts])[50]. In the first method, the brightest pixel in each row of the image was automatically selected using a Python script. Points were inspected and outlier coordinates, if present, were removed using one of three filtering approaches to enforce continuity of the patched membrane [https://github.com/BrohawnLab/TensionScripts]. A circle was fit to the coordinate list using the Python package circle-fit [https://pypi.org/project/circle-fit/]. In the second method, three points were manually chosen along the curve of the membrane (denoted $(x_1, y_1)$, $(x_2, y_2)$ and $(x_3, y_3)$). Patch radius was calculated from the coordinates using a Python script to solve the following equations:

$$A = x_1(y_2 - y_3) - y_1(x_2 - x_3) - x_2 y_3 - x_3 y_2 \tag{2}$$

$$B = (x_1^2 + y_1^2)(y_3 - y_2) + (x_2^2 + y_2^2)(y_1 - y_3) + (x_3^2 + y_3^2)(y_2 - y_1) \tag{3}$$

$$C = (x_1^2 + y_1^2)(x_2 - x_3) + (x_2^2 + y_2^2)(x_3 - x_1) + (x_3^2 + y_3^2)(x_1 - x_2) \tag{4}$$

$$D = (x_1^2 + y_1^2)(x_3 y_2 - x_2 y_3) + (x_2^2 + y_2^2)(x_1 y_3 - x_3 y_1) + (x_3^2 + y_3^2)(x_2 y_1 - x_1 y_2) \tag{5}$$

$$\text{radius} = \sqrt{\frac{B^2 + C^2 - 4AD}{4A^2}} \tag{6}$$

The value from *(6)*, the patch radius in pixels, was used to plot a circle on the image for validation. Images analyzed with both methods gave comparable results. Patch radii (r) were used to calculate membrane tension (*T*) during pressure (P) application using Laplace's law as previously described[19,20,24].

$$T = \Delta \Pr / 2 \tag{7}$$

**Reporting summary**

Further information on research design is available in the Nature Portfolio Reporting Summary linked to this article.

## Data availability

The data that support this study are available from the corresponding authors upon request. Source data are available with the manuscript. Source data are provided with this paper.

## Code availability

Our code used to quantify membrane tension[50] is available here [https://github.com/BrohawnLab].

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

## Acknowledgements
We thank Dr. E. Iscaoff, Dr. Y. Yu, A. Chou, and C. Stanley for providing *Xenopus* oocytes; Dr. E. Haswell for the MscS construct; Dr. E. Iscaoff and Dr. A. Winans for the GFP-CAAX construct; and members of the Brohawn laboratory, especially K. Tucker, Dr. D. Kern, and Dr. A. Elleman, for critical review and discussion. This work was supported by the New York Stem Cell Foundation (S.G.B.), NIH/NIGMS GM145869 (S.G.B.), and NIH/NINDS NS125102 (B.S.).

## Author contributions
B.S., T.A.D. and V.P. performed electrophysiology. R.A.R. contributed to the early stages of the project. T.A.D., B.S. and R.A.R. generated constructs. B.S. and T.A.D. analyzed data. T.A.D. generated figures and wrote scripts for data analysis. T.A.D. and S.G.B. wrote the paper with input from all authors. S.G.B. supervised the project.

## Competing interests
The authors declare no competing interests.
