## [Peer Review File · Nature Communications]

Tension activation of mechanosensitive two-pore domain K⁺ channels TRAAK, TREK-1, and TREK-2Reviewer #1 (Remarks to the Author):

In this study, the authors employed simultaneous patch imaging and recording to quantify the tension response of mechanosensitive potassium ion channels (TRAAK, TREK-1, and TREK-2). While acknowledging inherent limitations in tension sensitivity characterization, the authors used MscS, a well-characterized *E. coli* mechanosensitive channel, as their control, confirming the reliability of their approach. In their study, they found that TRAAK and TREK-1 exhibit sensitivity comparable to Piezo1 and MscS but with broader responses. TREK-2 responds differently with intermediate sensitivity. The authors calculated protein area changes and energy differences, supporting a structural model for channel opening. More interestingly, they explored ultrasound activation of mechanosensitive channels, finding that TRAAK and MscS respond similarly to ultrasound and pressure stimulation, suggesting both activate channels through membrane tension. Based on these findings, they propose that acoustic radiation force and streaming underlie ultrasound-induced mechanical activation. This study contributes insights into the molecular mechanisms of mechanosensitive channel activation and has implications for neuromodulation and sonogenetic applications.

While the study is important and well-conducted, there are certain points that require further explanation:

1. While you well characterized the tension response of ion channels TRAAK, TREK-1, and TREK-2, why did you only study the TRAAK channels in ultrasound stimulation? Is there a specific reason for doing so?
2. While you tried your best to minimize the limitations in tension sensitivity study, there is certainly some variability in resting tension, as you noted in the discussion. Do you think there is a better way to record the tension response, such as using imaging methods?
3. Is there a way to study the tension response under whole-cell circumstances? Do you think this is necessary for doing so?
4. There are various studies on ultrasound neuromodulation under different ultrasound simulation circumstances. Does your study support or oppose current findings? For example, explain the decrease in ultrasound-stimulated activity or the ultrasound-induced inhibitory effect. Please provide more discussion on the current findings.
5. Why did you choose this ultrasound parameter? Is it possible that your proposed tension mechanism fits for all ultrasound stimulation?
6. There is some confusion in your descriptions: In your Fig. 1 (B, D, F), you need to describe the meaning of each line more clearly.

Reviewer #2 (Remarks to the Author):

This is an interesting study to establish a quantitative analysis to measure the membrane tension and determine the tension responses of mechanosensitive ion channels TRAAK, TREK-1, TREK-2, and MscS. By simultaneous patch-clamp recording and fluorescent imaging, the authors have thus revealed membrane tension, rather than applied pressure, being the stimulus that promotes K2P (and MscS) channel opening. This study provides a new direction to advance our understanding of the biophysics properties of mechanosensitive ion channels and how these channels responding to ultrasound stimulus. A potential sonogenetics application is expected. Still, some concerns about the technology should be clarified.

1. It is important to show the membrane tension measurement is repeatable and with good reproducibility in the same patch-clamp recording.
2. The authors mentioned their efforts to minimize variability by pulling patches with low basal curvature and low resting channel activity. This is very important. Please clarify the details of how this can be achieved.
3. What are the limitations for the low tension measurement of negative pressure and ultrasound?
4. How were the thresholds to open mechanosensitive ion channels determined?

5. It would be helpful to show a correlation figure between ultrasound energy, ultrasound output, and membrane tension.
6. The data of Supplemental Figure 2A is confusing. It is surprised to see all 6 patches were 2.6 mN/m.
7. In Figure 2F&G, please provide the n numbers and how the statistic analyses were performed.
8. In Figure 4, apart from the pair recordings of ultrasound-pressure, please also show ultrasound-ultrasound, pressure-ultrasound, pressure-pressure. It is important to validate the data reproducibility.

POINT BY POINT RESPONSE

REVIEWER COMMENTS

Reviewer #1 (Remarks to the Author):

In this study, the authors employed simultaneous patch imaging and recording to quantify the tension response of mechanosensitive potassium ion channels (TRAAK, TREK-1, and TREK-2). While acknowledging inherent limitations in tension sensitivity characterization, the authors used MscS, a well-characterized *E. coli* mechanosensitive channel, as their control, confirming the reliability of their approach. In their study, they found that TRAAK and TREK-1 exhibit sensitivity comparable to Piezo1 and MscS but with broader responses. TREK-2 responds differently with intermediate sensitivity. The authors calculated protein area changes and energy differences, supporting a structural model for channel opening. More interestingly, they explored ultrasound activation of mechanosensitive channels, finding that TRAAK and MscS respond similarly to ultrasound and pressure stimulation, suggesting both activate channels through membrane tension. Based on these findings, they propose that acoustic radiation force and streaming underlie ultrasound-induced mechanical activation. This study contributes insights into the molecular mechanisms of mechanosensitive channel activation and has implications for neuromodulation and sonogenetic applications.

While the study is important and well-conducted, there are certain points that require further explanation:

1. While you well characterized the tension response of ion channels TRAAK, TREK-1, and TREK-2, why did you only study the TRAAK channels in ultrasound stimulation? Is there a specific reason for doing so?

Our preliminary data suggested TRAAK and MscS were most sensitive to low tension. We reasoned they would be most amenable to study in patch clamp experiments with ultrasound and pressure in which achieving high tensions prior to patch loss is challenging due to repeated mechanical stimuli. We also reasoned a higher degree of activation by low tension would make these channels more suitable as potential tools in sonogenetic applications.

2. While you tried your best to minimize the limitations in tension sensitivity study, there is certainly some variability in resting tension, as you noted in the discussion. Do you think there is a better way to record the tension response, such as using imaging methods?

We agree this is a limitation, but are unaware of a better way to record the tension response of channels. An optical reporter of tension has been reported (FlipTR (Fluorescent LIPid Tension Reporter), Colom et al. Nat. Chem. 2018), but this requires fluorescence lifetime imaging that we are not currently able to perform during patch clamp experiments. In addition, we would need to control for the probe altering channel properties as it intercalates into lipid bilayers and has not been used in studies of any mechanosensitive channel properties to date. We decided our approach was best suited for these first characterizations and comparisons of mechanosensitive K₂P tension response.

3. Is there a way to study the tension response under whole-cell circumstances? Do you think this is necessary for doing so?

This would be an interesting direction for future exploration. While theoretically possible to measure tension during whole-cell recordings using optical or magnetic tweezers that measure the force required to pull a tube of membrane away from the cell, these experiments are likely to be challenging to establish and we are not currently equipped to perform them.

4. There are various studies on ultrasound neuromodulation under different ultrasound simulation circumstances. Does your study support or oppose current findings? For example, explain the decrease in ultrasound-stimulated activity or the ultrasound-induced inhibitory effect. Please provide more discussion on the current findings.

We have added additional text to this point in the discussion. We believe our work suggests activation of mechanosensitive channels including K2Ps could underlie some reports of ultrasonic modulation of neuronal activity. However, we prefer not to try and interpret any specific result at this point given the extremely wide range of targets, stimulation parameters, and effects reported. In addition, the effects of ultrasonically activating TRAAK and TREK-1 in nodes of Ranvier within myelinated axons could vary depending on properties of different neurons. Increased potassium conductance could inhibit spiking by hyperpolarizing cells and decreasing input resistance. Alternatively, it could stimulate or facilitate spiking under conditions of limited voltage-gated sodium channel availability since hyperpolarization promotes Na_v recovery from inactivation.

5. Why did you choose this ultrasound parameter? Is it possible that your proposed tension mechanism fits for all ultrasound stimulation?

We chose a 3.5 MHz transducer as a representative for low frequency stimulation. We have added additional data in Supplementary Fig. 3 showing TRAAK is similarly activated by lower frequency (2.25 MHz) stimulation. Together with our previous report using 5 MHz stimulation (Sorum et al. PNAS 2021), these data show TRAAK is readily activated over this frequency band. We have not further explored the large possible parameter space for ultrasound stimulation.

6. There is some confusion in your descriptions: In your Fig. 1 (B, D, F), you need to describe the meaning of each line more clearly.

We thank the reviewer for pointing out this ambiguity and have clarified the text.

Reviewer #2 (Remarks to the Author):

This is an interesting study to establish a quantitative analysis to measure the membrane tension and determine the tension responses of mechanosensitive ion channels TRAAK, TREK-1, TREK-2, and MscS. By simultaneous patch-clamp recording and fluorescent imaging, the authors have thus revealed membrane tension, rather than applied pressure, being the stimulus that promotes K2P (and MscS) channel opening. This study provides a new direction to advance our understanding of the biophysics properties of mechanosensitive ion channels and how these channels responding to ultrasound stimulus. A potential sonogenetics application is expected. Still, some concerns about the technology should be clarified.

1. It is important to show the membrane tension measurement is repeatable and with good reproducibility in the same patch-clamp recording.

We have added data to show good reproducibility of the tension measurement within a patch in Supplementary Figure 6A. Paired pressure steps from the same patch that result in comparable channel activation result in statistically indistinguishable tension.

2. The authors mentioned their efforts to minimize variability by pulling patches with low basal curvature and low resting channel activity. This is very important. Please clarify the details of how this can be achieved.

We have expanded on this point in the Methods. This was done empirically since we were not able to measure resting tension. "We attempted to minimize variability in resting tension by pulling patches with low basal curvature and low resting channel activity consistent with low basal tension. Pipettes pulled with short taper and moderate diameter tips (with a corresponding resistance of 2-3 M Ω in recording solutions) resulted in a higher proportion of patches with low basal activity. This also reduced frequency of run up of basal activity (likely due to patch creep up the pipette wall⁴⁵) after repeated mechanical stimuli. Data from patches with basal activity corresponding to $I/I_{\text{max}} > 0.25$ was not analyzed."

3. What are the limitations for the low tension measurement of negative pressure and ultrasound?

There are two limitations to measuring low tension. First, as noted above, there is resting tension in patches that we try to minimize, but do not measure. Second, it becomes increasingly challenging to calculate tension in very low-tension membranes with nearly infinite curvature. If the patch is flat (with an infinite radius of

curvature) rather than hemispherical, calculated tension from the Laplace-Young equation ($T = \Delta Pr/2$) is not meaningful. We were able to measure tension as low as 0.36 mN/m in our experiments.

4. How were the thresholds to open mechanosensitive ion channels determined?

We define “threshold” as tension required for 10% activation from fits to I/I_{\max} versus tension data.

5. It would be helpful to show a correlation figure between ultrasound energy, ultrasound output, and membrane tension.

We have added a new Supplementary Figure 3 quantifying ultrasound power and channel TRAAK activation.

6. The data of Supplemental Figure 2A is confusing. It is surprised to see all 6 patches were 2.6 mN/m.

We thank the reviewer for noting this and apologize for the error. We have corrected these values.

7. In Figure 2F&G, please provide the n numbers and how the statistic analyses were performed.

Added.

8. In Figure 4, apart from the pair recordings of ultrasound-pressure, please also show ultrasound-ultrasound, pressure-ultrasound, pressure-pressure. It is important to validate the data reproducibility.

We have added these controls in a new Supplementary Figure 6. Paired pressure, ultrasound, and ultrasound preceding pressure stimuli from the same patch that result in comparable TRAAK activation show indistinguishable curvature and tension.

Reviewer #1 (Remarks to the Author):

Appreciation is extended to the authors for conscientiously considering the review comments and making appropriate adjustments based on the provided suggestions. However, there remain some concerns that require clarification.

Q1: In your discussion on page 1, you mention, "Ultrasound has been shown to produce both inhibitory and excitatory effects on neuronal activity depending on the system and stimulation parameters used²⁶." I don't think the reference provided supports for this assertion. Please verify. Additionally, kindly provide the reference that supports your subsequent assertion, "Our results suggest activation of mechanosensitive channels including K2Ps could contribute to some of these effects and that targeting myelinated fibers, where TRAAK and TREK-1 reside in nodes of Ranvier, could modulate neuronal spiking by increasing nodal potassium conductance."

Q2: In your study, you apply *Xenopus laevis* oocytes to test if the membrane tension is responsible for TRAAK, TREK-1, and TREK-2 ion channels (Fig2). Regarding the tension is related to the cell property, more powerful support of your finding can be made by applying more cells.

Q3: In your Fig5 you compare tension response ranges of mechanosensitive channels TRAAK, TREK-1, TREK-2, MscS (from this study), Piezo1 (from ref.13), and MscL (from ref.11). Piezo1 showed a largest tension response, it would be much interesting to see the tension and current changes under ultrasound stimulation.

Q4: In your Fig5 you compare tension response ranges of mechanosensitive channels TRAAK, TREK-1, TREK-2, MscS (from this study), Piezo1 (from ref.13), and MscL (from ref.11). MscL showed a smallest tension response response. In Tenter's and Sun's lab, the sonogenetics showed a considerable effect of sonogenetics, can you make a comment or discussion on that.

Q5: While you mentioned in your response to Q4, "However, we prefer not to try and interpret any specific result at this point given the extremely wide range of targets, stimulation parameters, and effects reported," I strongly suggest that you offer a more insightful discussion. For example, 1 compare your current findings with the current mechanism proposed by different labs and why your work is important; for aspect 2 compare your work with either current findings of ultrasound neuromodulation or sonogenetics.

Reviewer #2 (Remarks to the Author):

The authors have addressed most of my comments. I have a few minor points.

1. It would be informative to provide the lowest membrane tension (0.36 mN/m) can be measured in the result of discussion section.
2. The paired recordings of pressure preceding ultrasound stimuli are missing (Supplementary Figure 6).
3. Please correct in the Figure 1 legend. It should be (B,D,F) but not (B,D,E).

Reviewer #2 (Remarks on code availability):

I am capable of evaluating the code.

Point by point response

Reviewer #1 (Remarks to the Author):

Appreciation is extended to the authors for conscientiously considering the review comments and making appropriate adjustments based on the provided suggestions. However, there remain some concerns that require clarification.

Q1: In your discussion on page 1, you mention, "Ultrasound has been shown to produce both inhibitory and excitatory effects on neuronal activity depending on the system and stimulation parameters used²⁶." I don't think the reference provided supports for this assertion. Please verify. Additionally, kindly provide the reference that supports your subsequent assertion, "Our results suggest activation of mechanosensitive channels including K2Ps could contribute to some of these effects and that targeting myelinated fibers, where TRAAK and TREK-1 reside in nodes of Ranvier, could modulate neuronal spiking by increasing nodal potassium conductance."

Thank you for noting this error. The first reference has been corrected from (26) to (27).

Blackmore et al. Ultrasound Neuromodulation: A Review of Results, Mechanisms and Safety. *Ultrasound in Medicine & Biology* **45**, 1509–1536 (2019).

We have reworded the subsequent sentence and added citations to relevant references (20,21) that reported localization of TRAAK and TREK-1 to nodes of Ranvier in myelinated axons and their functional contribution to spike propagation.

Q2: In your study, you apply *Xenopus laevis* oocytes to test if the membrane tension is responsible for TRAAK, TREK-1, and TREK-2 ion channels (Fig2). Regrading the tension is related to the cell property, more powerful support of your finding can be made by applying more cells.

All aggregated data include recordings made from $n \geq 2$ cells (i.e. Figures 1B,D,F; 2B-G; 4C; Supplementary Figures 1B-D; 3A-F; 4A-F; 5C; 6A-C). No relevant differences were observed between patches from different cells. We added a statement in the Methods to this effect and added the number of cells used in tension calculations to the Figure legends. We further note that experiments to calculate tension were performed in excised patches in which membrane tension is only due to in-plane tension in the lipid bilayer, a cell extrinsic property. This differs from an intact cell, where membrane tension is due to a combination of in-plane lipid bilayer tension and membrane-cortical cytoskeleton attachments^{1,2}.

1. Belly, H. D. *et al.* Cell protrusions and contractions generate long-range membrane tension propagation. *Cell* **186**, 3049-3061.e15 (2023).

2. Sitarska, E. & Diz-Muñoz, A. Pay attention to membrane tension: Mechanobiology of the cell surface. *Curr. Opin. Cell Biol.* **66**, 11–18 (2020).

Q3: In your Fig5 you compare tension response ranges of mechanosensitive channels TRAAK, TREK-1, TREK-2, MscS (from this study), Piezo1 (from ref.13), and MscL (from ref.11). Piezo1 showed a largest tension response, it would be much interesting to see the tension and current changes under ultrasound stimulation.

We agree this is an interesting future direction, but it is outside the intended scope of this study in which we characterize mechanosensitive K2P channels. Ultrasound activation of Piezo1 has been demonstrated by others as noted in the discussion (references 31,39,40).

Q4: In your Fig5 you compare tension response ranges of mechanosensitive channels TRAAK, TREK-1, TREK-2, MscS (from this study), Piezo1 (from ref.13), and MscL (from ref.11). MscL showed a smallest tension response. In Tenter's and Sun's lab, the sonogenetics showed a considerable effect of sonogetics, can you make a comment or discussion on that.

Thank you for the suggestion. We have added references and text to the discussion regarding the use of tension-sensitized MscL mutants for sonogenetics by these groups.

Q5: While you mentioned in your response to Q4, "However, we prefer not to try and interpret any specific result at this point given the extremely wide range of targets, stimulation parameters, and effects reported," I strongly suggest that you offer a more insightful discussion. For example, 1 compare your current findings with the current mechanism proposed by different labs and why your work is important; for aspect 2 compare your work with either current findings of ultrasound neuromodulation or sonogenetics.

We have added text to the discussion related to these points.

Reviewer #2 (Remarks to the Author):

The authors have addressed most of my comments. I have a few minor points.

1. It would be informative to provide the lowest membrane tension (0.36 mN/m) can be measured in the result of discussion section.

Added.

2. The paired recordings of pressure preceding ultrasound stimuli are missing (Supplementary Figure 6).

Thank you for pointing out the ambiguity in our presentation. Paired recordings of pressure preceding ultrasound stimuli were presented in Figure 4C. Stimuli order is now stated explicitly in the legends of Figure 4C (pressure precedes ultrasound) and Supplementary Figure 6 (ultrasound precedes pressure).

3. Please correct in the Figure 1 legend. It should be (B,D,F) but not (B,D,E).

Corrected. Thank you for noting this error.